# Structural and Functional Remodeling of Mitochondria in Cardiac Diseases

**DOI:** 10.3390/ijms22084167

**Published:** 2021-04-17

**Authors:** Xiaonan Sun, Jalen Alford, Hongyu Qiu

**Affiliations:** Center for Molecular and Translational Medicine, Institute of Biomedical Science, Georgia State University, Atlanta, GA 30303, USA; xsun13@gsu.edu (X.S.); jalford22@student.gsu.edu (J.A.)

**Keywords:** mitochondria remodeling, heart disease, metabolism, heart failure

## Abstract

Mitochondria undergo structural and functional remodeling to meet the cell demand in response to the intracellular and extracellular stimulations, playing an essential role in maintaining normal cellular function. Merging evidence demonstrated that dysregulation of mitochondrial remodeling is a fundamental driving force of complex human diseases, highlighting its crucial pathophysiological roles and therapeutic potential. In this review, we outlined the progress of the molecular basis of mitochondrial structural and functional remodeling and their regulatory network. In particular, we summarized the latest evidence of the fundamental association of impaired mitochondrial remodeling in developing diverse cardiac diseases and the underlying mechanisms. We also explored the therapeutic potential related to mitochondrial remodeling and future research direction. This updated information would improve our knowledge of mitochondrial biology and cardiac diseases’ pathogenesis, which would inspire new potential strategies for treating these diseases by targeting mitochondria remodeling.

## 1. Introduction

Mitochondria is a double-membrane-bound organelle located in the cytoplasm of most eukaryotic cells. As a central energy station of the cells, mitochondria generate adenosine triphosphate (ATP) productions via oxidative phosphorylation (OXPHOS) to maintain the normal cellular metabolic homeostasis and play a critical role in normal cell functions [1]. Mitochondria also exhibit many different characters through regulating intracellular calcium (Ca^2+^) homeostasis, reactive oxygen species (ROS) generation, and cell death and survival pathway, and thus, control the cell fates under stress [2,3].

As a highly dynamic and responsive organelle, mitochondria can be adapted by both structural and functional remodeling to meet the cell demand in response to the intracellular and extracellular stimulations. The structural remodeling of mitochondria includes the changes in mitochondrial morphology, number, and distribution within the cell through multiple processes, such as fission, fusion, mitophagy and biogenesis, shape transition, and positioning. Additionally, mitochondria form a complex interconnected network within the cell and undergo a functional remodeling in response to diverse cellular pathways, such as metabolism, intracellular Ca^2+^ signaling, apoptosis, mitosis, and mitochondrial DNA replication, to ensure a well-coordinated response to environmental stresses [4].

Unlike other organelles, mitochondria have their own replication mitochondrial DNA (mt-DNA or mtDNA), which can encode the electron transport chain (ETC) components and other RNAs [5]. Mutation of mitochondrial genes will cause mitochondrial dysfunction and monogenic syndromes, such as Leigh’s disease and MELAS syndrome, characterized by mitochondrial myopathy, encephalopathy lactic acidosis, and stroke-like episodes [6]. Besides, the mitochondrial structure and function are highly regulated by the nuclear-encoded proteins. It has been shown that mitochondrial remodeling plays an essential role in the pathogenesis of diverse diseases, including cardiovascular diseases, metabolic disorders, neurological diseases [7,8,9]. Among these pathological conditions, mitochondrial alterations may be either a primary mechanism due to mutations in mitochondrial genes or a secondary process caused by the regulating network.

The heart is a high-energy-demanding organ, and its function largely relies on the ATP produced in mitochondria. Merging evidence demonstrated that mitochondrial dysfunction is the fundamental driving force of the various cardiac diseases despite the diversity of the primary causes, highlighting the importance of understanding the mitochondrial remodeling mechanisms in the heart [10,11]. In this review, we outlined the progress of the molecular basis in both mitochondrial structural and functional remodeling. In particular, we summarized the latest evidence of mitochondrial remodeling in developing different cardiac diseases and highlighted the underlying mechanisms. We also explored the therapeutical potential related to mitochondrial remodeling and associated future research direction. This updated information would improve our knowledge of mitochondrial biology and cardiac diseases’ pathogenesis, which would inspire new potential strategies for treating these diseases by targeting mitochondria remodeling.

## 2. Mitochondrial Remodeling in Cardiac Diseases

Cardiac diseases have become the primary cause of mortality and morbidity in most countries. Heart failure (HF) is a well-known typical last stage of different cardiac diseases. With the extensive studies from basic science to clinical research, the fundamental mechanisms of HF development are still not fully understood [12]. The heart is one of the highest energy-demanded organs in the human body that its function depends on ATP synthesis by oxidative metabolism in mitochondria. Thus, cardiomyocytes are uniquely sensitive to mitochondrial functional alterations. Increasing evidence indicates that, although the primary cause may be different, one of the central processes linking to various cardiac diseases is the impairment of mitochondrial structural and functional remodeling. These abnormal processes are a driving force of cardiac diseases’ pathogenesis, impairing the cardiomyocyte function and survival, leading to HF development [13]. Here, we summarized the latest evidence of mitochondrial remodeling and metabolic changes in different cardiac diseases, emphasizing new potential strategies for the clinical study.

### 2.1. The Overview of the Molecular Bases in Cardiac Mitochondrial Remodeling

Cardiac mitochondria are a highly mobile organelle that can undergo dynamic alterations depending on the cellular demand, which subsequently changes the cellular capabilities and functions. To better understand the mechanisms involved in cardiac diseases, we briefly outlined the molecular bases associated with cardiac mitochondrial structural and functional remodeling.

#### 2.1.1. Mitochondrial Structural Remodeling

Although mitochondrial morphologic changes may be less in cardiomyocytes than many other cell types, evidence indicated that cardiac mitochondria maintain the function through dynamic altering their size, number, and shape in response to the intercellular environments. These mitochondrial dynamics are essential for cellular homeostasis in adult myocytes under physiological conditions. Mitochondria continuously divide by the process of fission and merge by the process of fusion. They can also undergo a mitochondrial shape transition (MiST) between rounded and elongated mitochondrial morphologies independent of fission/fusion. These processes are positively related to mitochondrial mitophagy and biogenesis. Besides, mitochondrial morphology can be controlled by the interactions with the cytoskeleton and the endoplasmic reticulum (ER).

Mitochondrial fission is a process that refers to the division of mitochondria at the inner and outer membranes resulting in a single mitochondrion into two mitochondria. At the same time, fusion is another essential process for maintaining mitochondrial homogenates, which contains a fusion of the outer and inner mitochondrial membranes to create fewer but larger mitochondria. It is known that mitochondrial fission machinery is mainly regulated by dynamin-related protein 1 (DRP1), a member of the dynamin superfamily [14]. The fission process begins from the translocation of DRP1 from the cytosol to the outer mitochondrial membrane (OMM), where DRP1 interacts with a few OMM proteins, including mitochondrial fission protein 1 (Fis1), the mitochondrial fission factor (MFF), mitochondrial dynamics proteins of 51 and 49 kDa (MiD51 and MiD49), which are also known as mitochondrial elongation factors 1 and 2 (MIEF1 and 2) [15], forming a complex which mediates fission through wrap-around and constricts the mitochondrial tubule [16]. Some studies have shown that the overexpression of Fis1 induces mitochondrial fragmentation, while deficiency of Fis1 results in mitochondrial elongation [17]. Reciprocally, the silencing MFF increases mitochondrial fusion, whereas its overexpression leads to mitochondrial fragmentation, which could also be induced by external stimuli, such as by sarco-/endoplasmic ATPase inhibitors [18,19,20]. On the other hand, mitochondria fusion is mediated by several different dynamin-related large GTPases, mainly include mitofusin-1 (MFN1), mitofusin-2 (MFN2), optic atrophy protein-1 (OPA1), and also MIEF1 [21,22,23]. While MFN1/2 catalyzes the outer membrane fusion, OPA1 regulates the fusion of the inner membrane [24]. In addition, a recent study found that two forms of OPA1 cooperate to complete fusion of the mitochondrial inner-membrane, as the long-form of OPA1(L-OPA1) mainly mediates the membrane docking and lipid mixing, the short- form of OPA1 works together with L-OPA1 for mediating membrane pore opening [24].

Although mitochondrial fission and fusion are essential for the dynamic and structure change of mitochondria, studies show the MiST can occur independently of the fission and fusion process. It has been noticed that the MiST is vital to mitochondrial function, e.g., the long and tubular mitochondria are more active in producing ATP, while the round and short mitochondria are more related to cell dividing and pathological conditions [25,26]. Multiple cellular signals may be involved in MiST. Researchers demonstrate that MiST is associated with the actin polymerization induced by the increased level of cytosolic Ca^2+^. This process is governed by the mitochondrial protein mitochondrial Rho GTPase 1 (Miro1), a transmembrane protein consisting of two GTPase domains and two EF-hands, EF1 and EF2 (helix-loop-helix structural domain found in calcium-binding proteins) [27]. A recent study found that the distinct mechanisms regulating actin polymerization and depolarization are temporally associated with MiST [28,29]. Moreover, mitochondrial morphology and dynamics are partially regulated by the ER membrane contact sites (MCSs). ER MCSs stipulate the position of mitochondrial constriction and fission [30]. It has also been shown that mitochondrial fusion machinery could accumulate and assemble at ER MCSs. The hotspots of ER MCSs, where the mitochondrial fission and fusion occur, enable a rapid response of metabolic change [31].

The maintenance of mitochondrial number is a highly dynamic process that includes the synthesis of new mitochondrial components (biogenesis) and the removal of dysfunctional or old mitochondria (mitophagy). The increasing of mitochondria requires the protein expression from both mitochondrial DNA (mtDNA) and nuclear genome and requires cross-talk between them. Mitochondrial biogenesis is highly regulated by the vast of genome-encoded transcriptional elements, such as peroxisome proliferator-activated receptor γ (PPARγ) coactivator 1α (PGC-1α), estrogen-related receptor (ERR), nuclear respiratory factors (NRFs) [32,33]. Under energy depletion or cell growth, PGC-1α can be activated, stimulating the increase in mitochondria and oxidative metabolic [34]. Mechanistically, the activation of PGC-1α results in the induction of NRFs and ERR gene expression, regulating the mitochondrial genome’s transcription, like the mitochondrial transcription factor A (TFAM) [35]. On the contrast, mitochondrial mitophagy is the primary mechanism to reduce the mitochondrial number by moving the damaged mitochondria in the cell [36]. The most well-studied mitophagy mechanism is through a pathway involving the PINK1 (PTEN-induced kinase 1), an outer mitochondrial membrane kinase, and Parkin, an E3 ubiquitin ligase [37]. PINK1 is aggregated in the OMM, recruits Parkin translocated from the cytosol to mitochondria, and promoted ubiquitination [38]. Another signaling pathway of the NIP3-like protein X (NIX)/Bcl-2 interacting protein 3 (BNIP3) is also involved in the selective mitochondrial clearance. In BNIP3/Nix double knockout heart, mitochondrial morphology and function are notably disturbed [39]. Cardiac mitochondrial biogenesis has been demonstrated in chronic exercise conditioning, and mitochondrial fission is considered a component of the normal adaptation to increased energetic demand during exercise to mitochondrial function [40].

Although the mechanistic details of each process have yet still not been fully understood, recent work has helped define the molecular dynamics and the network of fission, fusion, shape transition, biogenesis, and mitophagy. These works have been well-reviewed recently [4,41]. The intersection of these pathways, with varied effects on mitochondrial morphology, further indicates the morphological complexity of mitochondria. A summary picture of the mitochondrial dynamic changes is illustrated in Figure 1.

#### 2.1.2. Mitochondrial Functional Remodeling

Besides the structural remodeling mentioned above, cardiomyocytes are uniquely sensitive to mitochondrial functional dynamic alterations due to the high energy requirements of rhythmic contraction. For maintaining the metabolic and energy homeostasis, cardiac mitochondria undergo several types of dynamic functional remodeling, such as the shift of mitochondrial fuel selection, the modification of calcium (Ca^2+^) handling, and the rebalance of ROS production and antioxidant defense. As illustrated in Figure 2, these functional remodeling are regulated by diverse signaling and play critical roles in controlling cellular programs.

As the primary site for generating energy, mitochondria determine and select fuel utilization to meet the cell demand on different conditions. In most cells, glucose and fatty acids are the primary energy sources for mitochondria, although other fuel resources may also be used, such as amino acids and ketone. The shift between fuels provides metabolic flexibility in the cells. It has been shown that the pyruvate dehydrogenase (PDH) and PDH kinase (PDK) play critical roles in the fuel shift of mitochondria. For example, in the low energy condition, increased Acyl-CoA may inhibit PDH directly or via activating PDH kinase, which further phosphorylates and inhibits PDH. Thus, mitochondria metabolism shifts from glucose oxidation toward fatty acid oxidation. On the other hand, the dephosphorylates of PDH will restore glucose oxidation. It is also reported that fatty acid oxidation is regulated by the Malonyl-CoA, a derivation of CoA. Increased Malonyl-CoA will block fatty acid oxidation by inhibiting carnitine palmitoyltransferases-1 (CPT1), the leading shuttle for the mitochondrial fatty acid transfer [42]. It is also reported that, under some conditions, the metabolic process involves a competition of glucose and fatty acids (FAs) for substrate utilization, which is also known as the glucose-FA cycle [43]. It has been shown that fatty acid acts as ligands to activate PPARα, upregulating PDK expression and phosphorylation, which inactivates PDH to prevent pyruvate from entering the mitochondria, resulting in suppression of glucose oxidation. Simultaneously, acetyl-CoA and NADH produced during FA oxidation can upregulate PDK activity to produce similar effects [44].

Ca^2+^ uptake into the mitochondrial matrix is critical to cellular function due to its roles in producing ATP and in initiating cell death. Ca^2+^ can directly activate PDH phosphatase and the dehydrogenases of the tricarboxylic acid (TCA) circle [45]. An increase in mitochondrial Ca^2+^ level at a physiological range promotes TCA circle activity and increases ATP production [46]. However, excessive mitochondrial Ca^2+^ level induced by stress has been founded to be toxic, which results in the loss of the ability of mitochondria to generate ATP and mitochondrial permeability transition pore (mPTP) opening, leading to cell death. Mitochondria Ca^2+^ homeostasis is determined by the dynamic regulation of both mitochondrial Ca^2+^ influx and efflux. It is generally accepted that mitochondria regulate Ca^2+^ efflux via the mitochondrial Na^+^/Ca^2+^ (sodium–calcium ion) exchanger (NCX) in excitable cells and via mPTP opening that is usually associated with mitochondrial stress and apoptotic cell death. In contrast, the influx of mitochondrial Ca^2+^ is considered to be governed by the mitochondrial Ca^2+^ uniporter (MCU) complex [47]. However, the mechanisms involved in mitochondrial Ca^2+^ transport remain largely unclear. Serval studies disputed that Ca^2+^ uptake in cardiac mitochondria might be limited under the physiological condition due to the lower expression levels of the uniporter compared to other tissues [48,49]. It showed that mitochondria regulate Ca^2+^ homeostasis by communicating with the SR via mitochondria-associated ER membranes (MAMs) [50,51,52]. The function of ER-mitochondria interaction is regulated by several factors located on the MAMs, such as inositol 1,4,5-trisphosphate (IP3) receptor (IP3Rs), voltage-dependent anion-selective channel 1 (VDAC1), and glucose-regulated protein 75 (Grp75), plays a critical role in the ER-mitochondrial Ca^2+^ transfer [53]. Through MAMs, Ca^2+^ released from ER is captured by mitochondria via VDACs as well as mitochondrial Ca^2+^ uniporter (MCU) complex [54]. With the high demand for ATP production, cells generally increase the contact area between ER and mitochondria, which in turn increases the release of Ca^2+^ from the ER and causing Ca^2+^ to flow into the mitochondria [55]. Studies with Duchenne muscular dystrophy (DMD) models also assumed that the Ca^2+^ transport systems of cardiac mitochondria could be activated as an adaptation to stressful conditions or pathological models to compensate for the disruption of the ER. These activations, including the increased rate of both Ca^2+^ uptake and efflux, result in mitochondrial hyperfunction, which may contribute to the adaption to the cellular function under stress; however, it may also lead to hypertrophy-associated pathologies [56,57,58].

ROS is the byproduct of OXPHOS and mainly exists as superoxide, including oxygen, hydroxyl radicals, or peroxidases. Mitochondria is the primary source and target of ROS. The central part of ROS can be generated during the NADPH oxidase and mitochondrial electron transport [59]. Under normal conditions, a small amount of ROS is produced and will quickly undergo dismutation to hydrogen peroxide by superoxide dismutase. However, under some stress conditions, excessive ROS and superoxide production exceeds the antioxidant defense ability [60], becoming toxic to the cell.

Excessive ROS may damage the mitochondrial genome and protein, and mitophagy will be triggered. Due to mtDNA’s lower repairability, the lack of protective barrier histone-like protein, and the disability to transfer accumulated •O^2−^ out of the mitochondrial membrane, mitochondrial can be the significant damage target by ROS [61]. Moreover, with the accumulation of ROS, it can react with polyunsaturated fatty acids, which may induce lipid peroxidation. Hydroxyl-alkenes, like 4-hydroxy-2-nonenal (HNE), are the main product of lipid peroxidation. HNE can react with cysteine sulfhydryl groups, lysine amino groups, and histidine imidazole groups by forming covalent adducts on proteins. It also can react with other low-molecular-weight compounds, like glutathione and DAN [62]. Moreover, HNE is considered as the second messenger for the extent of oxidative stress [63]. Besides, mPTP opening can be triggered by HNE, which may lead to an increased release of mitochondrial cytochrome c and other factors and promote cell death [64] (Figure 3).

### 2.2. The Association of Mitochondrial Remodeling in Pathological Cardiac Conditions

#### 2.2.1. Aging-Induced Cardiac Deterioration

A major risk factor for prevalent cardiovascular disease and HF is age. During aging, cardiac structure and function were progressively deteriorated, leading to increased susceptibility to heart failure. With aging increased, the heart exhibits a decrease in the number of myocytes with a concomitant increase in each cardiomyocyte’s size and an increased accumulation of lipid and areas of fibrosis. Although our understanding of cardiac aging remains limited, researchers have shown that mitochondrial fission and fusion processes are altered, and the function of mitophagy is disturbed during aging. All of these lead to the impaired mitochondrial biogenesis [65]. The impaired mitophagy also activates the mitochondrion-mediated apoptotic signals, leading to cell apoptosis [66]. The aging heart exhibits a declined functional remodeling in mitochondria, leading to ROS accumulation, which results in a reduced tolerance for stress and increases cell loss [67]. Moreover, researchers also consider that the hyper-function of mitochondria is contributed to aging. During the cell senescence, high ROS level, mitochondrial morphological change, and elevated metabolic rate are observed [68,69].

A most recent study showed that an exacerbation of the nuclear factor-kappa B (NF-κB)/nucleotide-binding domain and leucine-rich repeat-containing protein 3 (NLRP3) pathways might be responsible for the declined mitochondrial remodeling in the aging heart. The results from this study showed that the absence of NLRP3 prevented age-related mitochondrial dynamic alterations in cardiac muscle with minimal effects in cardiac autophagy during aging, and mice showed less mitochondrial damage than wild-type animals [70].

#### 2.2.2. Diabetic Cardiomyopathy (DCM)

Diabetes mellitus (DM) has become one of the most common chronic diseases worldwide and become the primary etiology for metabolic heart disease [71]. In the United States, about 25% of people over 65 years old suffered from DM. There are two main types of DM, Type 1 DM (or T1D), which is caused by the autoimmune destruction of the insulin-secreting pancreatic B cells); and Type 2 DM (or T2D), which is resulted from impaired insulin secretion secondary to systemic insulin resistance [72]. The heart is one of the significant targets of pathological metabolic change, and DCM is considered one of the leading causes of death in DM patients [73]. DCM is characterized by left ventricle (LV) hypertrophy, diastolic dysfunction, and cardiomyocyte rarefaction [74].

Studies have shown that the alterations in cardiac mitochondrial remodeling and metabolism contribute to the progress of DCM [75,76]. Serval evidence indicates that mitophagy plays a central role in mitochondrial quality control in diabetic cardiomyopathy, and dysfunctional mitophagy is associated with lipotoxicity [77]. For example, in a high-fat diet-induced type 2 DM mouse model, cardiomyocytes show impaired mitophagy and lipid accumulation [78]. In the streptozotocin (STZ)-induced diabetic cardiomyopathy(DM) animal model, PINK1 and Parkin protein levels are decreased [79]. Researchers also found FUN14 domain containing 1 (Fundc1), an outer mitochondrial membrane protein, is associated with mitophagy and MAM regulation. In diabetic hearts, the suppression of Fundc1 induced by AMP-activated protein kinase may become a novel therapeutic target for DCM [80]. In the genetically induced diabetic animals and high-fat diet treated animal models, studies demonstrate that at the early stage of diabetic cardiac dysfunction, mitochondrial biogenesis occurs with the activation of PGC-1α expression. Still, the expression of PGC-1α decreased with the injured mitochondrial function [81,82]. In addition, deficient Ca^2+^ handling is also a critical factor for the development of cardiac contractility dysfunction in the DM. In the T1D and T2D hearts, improved mitochondrial Ca^2+^ handling in cardiomyocytes may enhance the metabolic activity, indicating that the regulation of mitochondrial Ca^2+^ may provide a novel therapeutic target in DCM [83].

Besides, there is a dramatic alteration in the mitochondrial functional remodeling in DCM. In the normal heart, 60% of energy is provided from fatty acid oxidation, 40% from other sources, like glucose, amino acids, and ketone bodies. In the DCM, increased fatty acid utilization occurs; it shows mitochondrial substrate utilization excessively shifts to fatty acid oxidation and harms the heart function [84]. Researchers found that the DM’s systemic insulin resistance may induce the increased fatty acid utilization and then cause heart dysfunction [85]. The underlying mechanism is considered the over-accumulation of fatty acid in the heart and the ROS caused by excessive fatty acid oxidation. Thus, the process of cardiac lipotoxicity and ROS will trigger apoptosis and cell death [86,87].

Moreover, fatty acids can be metabolized into diacylglycerol (DAG) and ceramides. These intermediates may further aggravate the cardiac insulin resistance [88,89]. In the rat DCM, the expression of Transcription Factor A (TFAM) and the activities of enzymes involved in OXPHOS were reduced, along with increased oxidative stress [90].

#### 2.2.3. Hypertrophic Heart Disease and Heart Failure

Cardiac hypertrophy is characterized as a thickening of ventricle walls, mostly the left ventricular hypertrophy (LVH), such as caused by hypertension, but it can also happen to the right ventricle (RV) hypertrophy. Concomitant heart failure is usually the final stage of this cardiac pathological process [91]. Although the underlying mechanism is complex and far from fully understood, several metabolic signaling pathways are related to mitochondrial remodeling and have been linked to the development of hypertrophic heart diseases and heart failure.

First, researchers found that pathological cardiac hypertrophy and left ventricular dysfunction were developed in the early stage in the PINK knockout mice [92]. Additionally, in human HF, PINK1 protein expression is notably decreased [92]. Since the PINK1/Parkin pathway plays an essential role in cardiac mitochondria’ quality control, these observations indicate the involvements of impaired mitophagy in pathological cardiac hypertrophy and dysfunction.

Secondly, PGC-1a/ERRs, the key transcriptional regulators of mitochondrial metabolism and biogenesis, have been observed to be decreased in the failing human left ventricular tissue [93]. Genetic deletion of PGC-1a in the mice shows cardiac dysfunction under the stress of pressure overload condition [94]. Adult mice with induced PGC-1α deletion also displayed lower respiratory capacity and decreased transcripts encoding fatty acid oxidation (FAO) factors [95]. In addition, cardiac hypertrophy has also been associated with reduced myocardial fatty acid utilization and shifted to glucose utilization [96,97]. During cardiac hypertrophy, the genes which express PGC-1α/PPAR-driven FAO is down-regulated [98]. Additionally, FAO rates and β-hydroxy acyl CoA dehydrogenase, a fatty acid oxidative enzyme, are reduced in the pressure overload cardiac hypertrophy [99]. In the murine arteriovenous fistula (AVF) model, the volume overload leads to ventricular dysfunction and mitochondrial ROS production through the decrease of NADH oxidase super complex activity [100]. Besides the ROS generation, in the failing heart, the relative contribution of glucose utilization to ATP production increases, but this change is not matched by an increase in pyruvate oxidation in mitochondria. Ketone bodies might become a more relevant source of energy in the failing myocardium. However, the free fatty acid (FFA) uptake is higher than the fatty acid oxidation, resulting in toxic lipid intermediates accumulation [101].

Third, in the animal model of compensated hypertrophy-induced HF, researchers have shown the FOXO3a (forkhead box O3a)-BNIP3 (B-cell lymphoma 2/adenovirus E1B 19kDa interacting protein 3) pathway is upregulated. Increased expression of BNIP3 participants in the mitochondrial fragmentation, Ca^2+^ overload, and oxidative capacity decline [102,103].

Fourth, studies showed that the increase in mitochondrial sirtuin-3, a member of the sirtuins’ family, can attenuate cardiac hypertrophy by decreasing oxidative stress [104]. Recent studies also showed that deficiency of a valosin-containing protein (VCP) in the heart is associated with the pressure overload-induced pathologic cardiac hypertrophy and heart failure, likely through a mechanism by the activation of oxidative stress and the increased ROS production [105,106].

#### 2.2.4. Acute Myocardial Infarction (AMI)

Acute myocardial infarction is a very familiar ischemic heart disease and the primary cause of high mortality. Myocardial ischemia-reperfusion (IR) is a pathological condition exemplified by AMI followed by the immediate coronary intervention restoring the blood supply. IR is characterized by two consequential damages, the first one caused by ischemia due to the loss of the coronary blood supply to the myocardium, while the second one caused by the reperfusion due to the restoration of blood supply, also called reperfusion injury [107]. Cardiac ischemia impaired OXPHOS because of oxygen deprivation, resulting in ATP depletion, as reperfusion caused excessive ROS production and the mPTP opening, leading to cell death.

Studies have shown that several alterations in mitochondrial structural remodeling are involved in the IR damage. For example, it has been shown that IR injury may cause mitochondrial morphology to change due to the alteration of intracellular arrangement and expression of cytoskeletal proteins, such as the intracellular arrangement of β tubulin II, a cytoskeletal protein localized in mitochondria [108]. Additionally, there has been an alteration of mitophagy in heart ischemia/reperfusion (I/R) injury. Studies show appropriate mitophagy is a cardioprotective response [109], and loss of PINK1 results in an impaired mitophagy and increase infarct size after I/R [110]. On the other hand, the mitochondrial protein FUN14 domain containing 1 (FUNDC1) is also linked with mitophagy as its overexpression induced an enhanced mitochondrial degradation [111]. In the ischemic heart, upregulated CK2α deactivates FUNDC1 leading to decreased mitochondrial receptor-mediated mitophagy [112]. It is found that sodium thiosulfate preconditioning (SIPC) treatment can increase the expression of PGC-1α mRNA, which also increases mitochondrial copy number [113]. Researchers also found DJ-1 plays a cardiac protective role in the IR injury through regulating DRP1 SUMOylation and attenuating undue mitochondrial fission [114].

In addition, IR injury also causes essential alterations in mitochondrial functional remodeling. For example, IR injury will cause ROS production, disturbing ATP concentration by influencing ion channels, calcium-release channels, and sarco-ER. Due to the impairment of the ATPase-dependent ion transportation, mitochondrial calcium levels are increased during ischemia [115,116]. During ischemia, total tissue Ca^2+^ does not change, whereas mitochondrial Ca^2+^ increases, suggesting a redistribution of intracellular Ca^2+^ rather than the massive net accumulation during reperfusion after prolonged ischemia. Although the route by which Ca^2+^ enters the cytoplasm is controversial, it appears that Ca^2+^ channels are not involved [117]. Uncontrolled Ca^2+^ entry might occur through leaks or membrane defects caused by the accumulation of toxic metabolites (e.g., lysophospholipids, oxyradicals) and mechanical factors [117,118]. Alternatively, Ca^2+^ overload could result from the impairment of specific pathways utilized for Ca^2+^ uptake and extrusion [118]. One possible mechanism is Ca^2+^ entry through the Na^+^/ Ca^2+^ exchanger, which is far from its normal equilibrium due to a rise in the intracellular Na^+^ concentration after a period of ischemia. Another suggested mechanism is inhibition of the by acidosis during ischemia followed by reactivation on reperfusion, causing an efflux of H^+^ and an influx of Na^+,^ which, in turn, stimulates Ca^2+^ entry on the Na^+^/Ca^2+^ exchanger [119]. Besides, studies also showed that overexpression of VCP in the heart protects IR-induced infarction [120]. It is shown that overexpressing VCP increased post-translational protein degradation of the mitochondrial Ca^2+^ uptake protein 1 (MICU1), an activator of the mitochondria Ca2+ uniporter (MCU), resulting in reduced mitochondrial calcium uptake, subsequently preventing mPTP opening and ATP depletion under the Ca^2+^ challenge [121]. Whatever the entry route, the rise in intracellular Ca^2+^ is accompanied by an increase in mitochondrial Ca^2+^, which impairs oxidative phosphorylation. The relationship between mitochondrial calcium transport and ATP synthesis previously described may be a key factor for cell survival during postischemic reperfusion [122,123]. With the increased ROS, the opening of mPTP is also a critical mechanism that may cause the dysfunction of mitochondrial membrane potential, disturbing ATP generation and leading to ischemia-reperfusion injury and cell death.

Furthermore, Bax/Bak-mediated mitochondrial outer membrane permeabilization (MOMP) can lead to caspase-dependent apoptosis or caspase-independent cell death due to the loss of mitochondrial function. Due to this pivotal role in deciding cell fate, deregulation of MOMP impacts many diseases and represents a fruitful site for therapeutic intervention [124]. Necrosis plays an essential role during IR injury. Myocardial necrosis is the process of cell death with rapid disruption of cellular membrane potential, which leads to cell swell, cytolysis, and inflammation [125]. The opening of mPTP is the key factor for necrosis in the cardiac I/R injury [126]. Researchers found the apoptosis repressor with caspase recruitment domain (ARC) can prevent myocardial necrosis by inhibiting the opening of mitochondrial mPTP [127]. This finding may provide a novel strategy for cardiac protection.

## 3. Conclusion, Clinical Potential, and Future Perspective

Mitochondria is the critical “energy engine” exhibiting essential structural and functional remodeling to adapt to the cell demand under the physiological condition. Dysregulation of mitochondrial remodeling is associated with various cardiac diseases and HF, including abnormal mitochondrial fission and fusion, defective mitochondrial mitophagy and biogenesis, disproportional fuel utilization and metabolic shift, aberrant mitochondrial Ca^2+^ handling, and excessive ROS productions. These impairments result in a deficiency of ATP production, mPTP opening, and activation of apoptotic signaling, leading to cardiomyocyte death and dysfunction. An overview of mitochondrial structure and functional remodeling in cardiac diseases is illustrated in Figure 4.

Many clinical potentials have been observed by targeting these pathological mitochondrial remodeling. A recent study showed that treatment with melatonin(N-acetyl-5-methoxytryptamine, aMT) recovered aging-induced alteration of mitochondrial dynamics and cardiac autophagy. Melatonin supplementation also had an anti-apoptotic action in addition to restoring Nrf2-antioxidant capacity and improving mitochondria ultrastructure altered by aging [70]. Besides, mitochondria transplantation is considered an effective treatment for preserving the contractility of ventricular hypertrophy. Studies show that the localized intramyocardial injection of autologous mitochondria can protect the right ventricular hypertrophy by decreasing apoptotic cardiomyocytes’ loss [128]. Nowadays, in ischemia-reperfusion injury, mitochondrial transplantation is becoming a novel therapy. The delivery of mitochondria through the coronary arteries resulted in their rapid integration and widespread distribution throughout the heart, replaced damaged mitochondria, and increased myocardial function [129,130]. Besides, the development of novel methods for mitochondria monitoring under physiological or pathological conditions is essential. Researchers recently developed more localization-specific with low toxicity fluorescent dyes and probes from the γaryl-substituted pentamethine family for mitochondrial morphology, dynamic, and structure studies [131,132].

On the other hand, researchers have been attracted to explore new therapeutical potentials by targeting the signaling pathways associated with mitochondrial functional remodeling. For example, researchers found the growth differentiation factor 11 (GDF11), a member of the TGF-β superfamily, can enhance the communication of SR and mitochondria and increase mitochondrial Ca^2+^ uptake in the cardiomyocyte hypertrophy [133], indicating a future target for the drug therapy. ROS production also becomes a critical therapeutic target. In recent years, several mitochondria-targeted antioxidants are made to reduce mitochondrial oxidative damage, especially the antioxidant compounds incorporating ubiquinone (MitoQ) or vitamin E (MitoVit E). Experiments show it can protect against mitochondrial dysfunction in heart failure induced by pressure overload [134]. Researchers show that SS31, an antioxidant located in the inner mitochondrial membrane, can decrease mitochondrial ROS production by eliminating ROS/oxygen-free radicals and protecting the myocardium from IR injury [135,136,137]. Researchers also show cyclosporin A (CsA), which can inhibit the opening of mitochondrial mPTP, is a novel therapeutic drug for myocardial IR injury [138]. VCP is also considered a crucial therapeutic target due to its various effects in mitochondrial respiration, calcium hemostatic, mPTP opening, and ROS production, as well as its critical protective roles in cardiac stress [106,120,121]. Since cardiac energy metabolism remodeling is a vital element in heart disease pathogenesis, pharmacological agents to modulate metabolic remodeling are also becoming a promising therapeutic strategy for the treatment of cardiac disease [97].

Although many studies have established the association between pathological mitochondrial remodeling and various cardiac diseases’ pathogenesis and revealed some clinical therapeutic potential, the fundamental mechanisms involved remain largely unknown. Future research needs to focus on the molecular mechanisms involved in regulating mitochondrial structural remodeling in the heart, such as the genes and proteins involved in the rapid dynamic regulations in reorganizing fission and fusion and the signaling promoting mitophagy and biogenesis under stress. These processes involved both nuclear-encoded genes (such as proteome) and mtDNA; thus, understanding the cross-talk between the mtDNA and nuclear genome would be particularly important. Besides, metabolic remodeling plays an essential role in the pathogenesis of metabolic cardiomyopathy. It is also involved in many other cardiac diseases, such as hypertrophic heart disease, ischemic heart diseases. More research needs to determine the common mechanism controlling the mitochondrial fuel selection and the shift of metabolism in the heart and explore the strategies to restore substrate availability and ATP production. Furthermore, mitochondrial calcium hemostasis is crucial for mitochondrial ATP production and also mPTP opening. One challenge in the mechanistic study may result from rapid changes in the relative ion channels’ structure and function during heart beating. Since mitochondrial calcium hemostasis regulation is spatiotemporal dependent, better models and techniques need to be developed to tracing these fast and dynamic alterations. Moreover, exploring the strategy to maintain the balance of ROS and antioxidants would improve the therapeutic approach in heart failure. Although the understanding of mitochondrial dysfunction in cardiovascular diseases has been increased notably in recent years, a more effective drug target is still in its early stages. Future investigations are still needed for transferring new findings to potential therapeutic approaches.

## Figures and Tables

**Figure 1 ijms-22-04167-f001:**
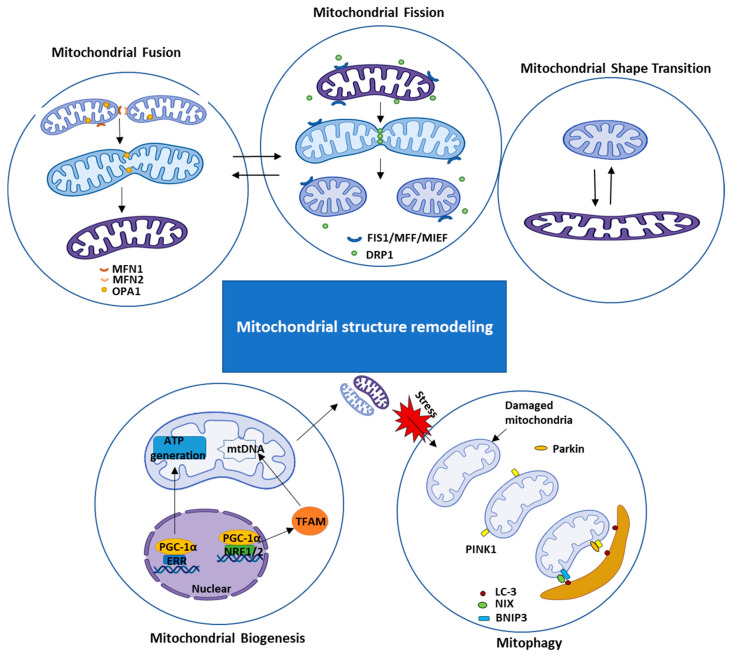
The Illustration of cardiac mitochondrial structure remodeling and mitochondrial dynamics. The mitochondria exhibit an adaption through changing mitochondrial morphology and numbers by fusion, fission, mitochondrial shape transition, and coordinated with mitochondrial biogenesis and mitophagy. Each process is mediated by several factors and also regulated by the interactions with other mitochondrial morphology changes. MFN1/2: mitofusin-1/2. OPA1: optic atrophy protein-1. Fis1: fission protein 1. MFF: mitochondrial fission factor. MIEF: mitochondrial elongation factors. DRP1: dynamin-related protein 1. LC-3: Microtubule-associated protein 1A/1B-light chain 3. NIX: NIP3-like protein X. BNIP3: Bcl-2 interacting protein 3. PGC1- α: peroxisome proliferator-activated receptor γ (PPARγ) coactivator 1α. ERR: estrogen-related receptor. NRFs: nuclear respiratory factors. TFAM: mitochondrial transcription factor A. PINK1: PTEN-induced kinase 1.

**Figure 2 ijms-22-04167-f002:**
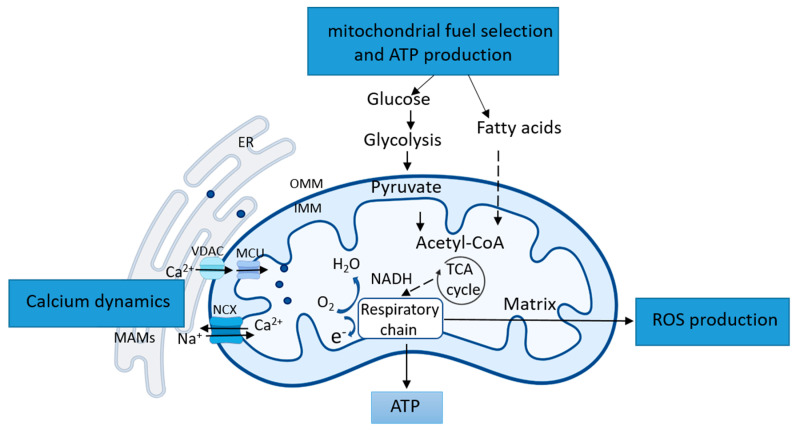
Overview of major mitochondrial functional remodeling. Mitochondria adapt to the cellular environment and stress to meet the energy demand under the various conditions through a serial of functional remodeling, such as the selection of fuel utilization and shift of metabolism, handling the dynamic calcium homeostasis and remaining the balance between reactive oxygen species (ROS) production and antioxidant defense. Multiple types of signaling are involved in these functional remodeling in mitochondria. TCA: tricarboxylic acid.OMM: outer mitochondrial membrane. IMM: inner mitochondrial membrane. ROS: reactive oxygen species. MCU: mitochondrial Ca^2+^ uniporter. VDAC: Voltage-dependent anion-selective channel. NCX: Na^+^/Ca^2+^ (sodium-calcium ion) exchanger. MAM: mitochondria-associated ER membranes.

**Figure 3 ijms-22-04167-f003:**
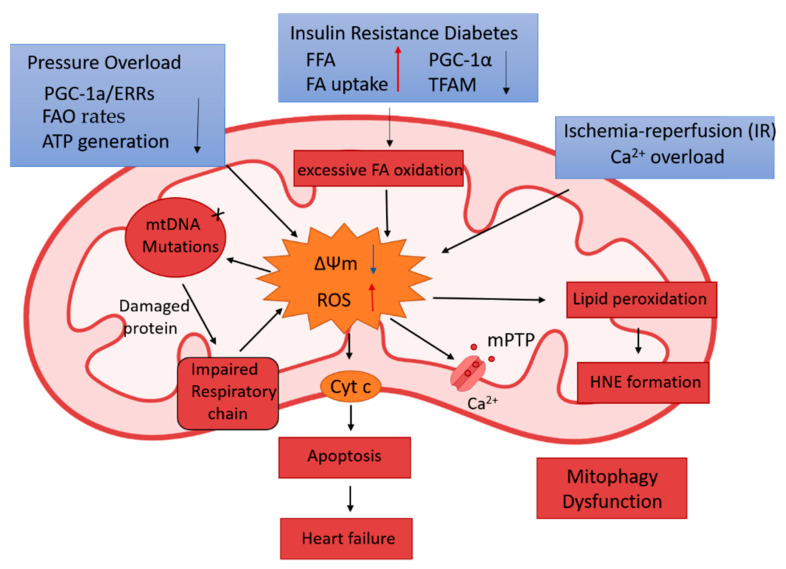
The production of ROS in mitochondria and its oxidative damage in different cardiac diseases. The increased ROS and the decreased mitochondrial membrane potential (∆Ψm) may lead to mtDNA mutation, impaired respiratory chain, lipid peroxidation, and mPTP opening, resulting in the release of cytochrome c (Cyt c), cell apoptosis, and heart failure. ROS: reactive oxygen species. mPTP: mitochondrial permeability transition pore. FFA: free fatty acid. FAO: fatty acid oxidation. Cyt C: cytochrome c. FA: fatty acid. TFAM: mitochondrial transcription factor A.

**Figure 4 ijms-22-04167-f004:**
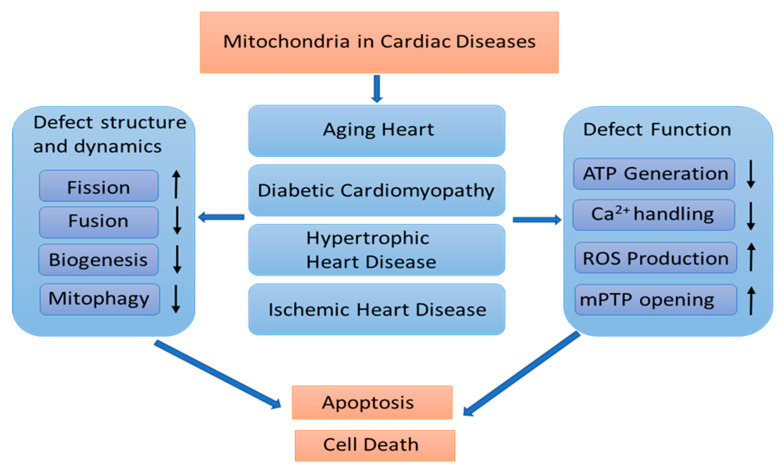
Overview of mitochondrial remodeling in cardiac diseases. In cardiac diseases, both mitochondrial structural and functional remodeling are changed, resulting in impaired mitochondrial function, leading to a declined ATP production, and ROS accumulation and mPTP opening, finally causing apoptosis and cell death. ROS: reactive oxygen species. mPTP: mitochondrial permeability transition pore.

## Data Availability

Not applicable.

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
