# Peer review of "Structural and Functional Remodeling of Mitochondria in Cardiac Diseases"

_ijms, 2021, doi:10.3390/ijms22084167_

Round 1
Reviewer 1 Report
A review by Sun et al. shows structural and functional remodeling of mitochondria in the development of various cardiac diseases. The work is logically described. It is also described in great detail. The latter is a problem. There are a lot of detailed reviews that describe the facts about the molecular bases and mechanisms of mitochondrial remodeling. I think that the authors should delete this part (chapter 2) and, after the introduction, focus their attention on the topic of the review, namely on the functional remodeling of mitochondria in cardiac diseases (chapter 3), where a brief explanation can be provided instead of describing in detail all mechanisms in a separate chapter. In addition, chapter 3 is best supplemented with several schematic figures describing the mechanisms of mitochondrial dysfunction in cardiac pathologies. It is also important in the sense that chapter 2 describes quite a lot of general well-known facts, and authors should show original references to them. In this case, the number of references for publications older than 5 years will increase significantly, this is unacceptable, a good modern review should contain adequate references to publications describing the latest trends in this area.
The authors point out the important role of mitochondria in the regulation of calcium homeostasis in heart cells. However, this is not quite true. See the papers PMID 25229137 and 23759742. Most cellular Ca2+ is in the ER. Cardiac mitochondria take up very minimal Ca2+ (~1% of total Ca2+ load per action potential) because expression levels of the uniporter are exceedingly low compared to other tissues. However, and this may be important for the authors, in pathologies associated with ER dysfunction, mitochondria can assume the main role in the regulation of calcium homeostasis, and thereby compensate for the disruption of the ER (PMID: 20971771; PMID: 25975620; PMID: 32569663 and others). It may also underlie the adaptation of the heart to pathological conditions. Moreover, in this case, mitochondrial hyperfunction may be observed, which the authors also need to discuss.
Reviewer 2 Report
The authors bring an updated review on the structural and functional remodeling of mitochondria in cardiac diseases.
The article is overall very well written and has a very good flow of the text, it was my pleasure to read it. However, I have few minor things which should be corrected:
A lot of abbreviations are not given in the first place they appear
Why are the following "Mitochondrial myopathy, Encephalopathy Lactic Acidosis, and Stroke-like episodes" starting with a capital letter?
Use enable instead of allows in the text, it has a slightly different meaning
Use a space between a number and a unit
Chapter 2.1 - you mentioned mitochondria + cytoskeleton interaction, what about mitochondria + endoplasmic reticulum - this should not be missing
Figure 1 - the text in the figure is too small, not readable. Also, explain all abbreviations from the figure (proteins) in the legend, the figure+caption should be understandable standalone
Line 117 - a verb is missing
Line 126 - after the phrase "leads to mitochondrial fragmentation" add "which could be also induced by external stimuli, such as by sarco-/endoplasmic ATPase inhibitors" and the following references should be added:
Sarco/Endoplasmic Reticulum Calcium ATPase Inhibitors: Beyond Anticancer Perspective. J Med Chem. 2020 Mar 12;63(5):1937-1963. doi: 10.1021/acs.jmedchem.9b01509.
Tailor-made fluorescent trilobolide to study its biological relevance. J Med Chem. 2014 Oct 9;57(19):7947-54. doi: 10.1021/jm500690j.
Line 156/157 - round mitochondria are related not only to pathological conditions but also to dividing cells, this information should be added, otherwise it is incomplete
Line 161 - "increased cytosolic calcium" is very vague term - it should be "increased levels (or concentrations) of cytosolic calcium ions" or something like that, not just calcium, etc.
The Ca2+ regulation of mitochondrial shape transition has been researched much more than it is outlined, and, thus, the authors should elaborate more on explaining this mechanism and regulation, it is only very briefly described and it deserved better explanation/discussion
a chapter on MAM (mitochondria-associated membranes) and the mitochondria-endoplasmic reticulum communication should be added, it is important and it is fully missing, it is a bit in chapter 2.7, but it is rather a mention, not discussed properly
the term "The maintenance of mitochondrial amount homeostasis" sounds weird, maybe just amount (without the homeostasis) would be sufficient, moreover, I also wonder, what do you mean by the amount, you should state it in the manuscript since the term amount and number of mitochondria is few times in the manuscript and the view on this strongly differs between researchers...
Chapter 2.6 - suddenly, you started using mDNA instead of mtDNA, which was used before in the text, it must be consistent, also check figure 3, mtDNA is a much more commonly used term
Line 307 "hemostasis" is probably a typo, used even further
"increase in" not increase of...
cytochrome c - c should not be capitalized
Figure 3 - explain the abbreviations in the legend
Also, in the article, the authors speak about mitochondria playing role in mitophagy and apoptosis but skip other types of cell death that mitochondria play role in and that can be relevant to the discussed diseases... it should be complete
Conclusion: a note about the importance of tool development for mitochondria monitoring in health and disease should be made, the following probes for such purpose should be referred:
Rational design of chemical ligands for selective mitochondrial targeting. Bioconjug Chem. 2013 Sep 18;24(9):1445-54. doi: 10.1021/bc400291f.
Highly selective mitochondrial probes based on fluorinated pentamethinium
salts: On two-photon properties and microscopic applications. Dyes and Pigments. Vol. 172, 2020, 107802. DOI: 10.1016/j.dyepig.2019.107802
Figure 4 - explain abbreviations from the legend as well as from the figure in the legend
Round 2
Reviewer 1 Report
I am satisfied with the changes made by the authors
Reviewer 2 Report
The authors have significantly improved the manuscript and included comments and suggestions raised by the reviewer. I recommend accepting the article.